# The development of new occupation practitioners in China's first-tier cities: A comparative analysis

**Yuxiang Zhang**, **Anhang Chen** *, **Linzhen Li**, **Huiqin Zhang**

College of Management Science, Chengdu University of Technology, Chengdu, Sichuan, P. R. China

* chenanhang1997@163.com

**Data Availability Statement:** All relevant data are within the paper and its Supporting Information files.

**Funding:** This research was funded by the Soft science Project of Sichuan Science and Technology

## Abstract

Owing to the increasingly complex economic environment and difficult employment situation, a large number of new occupations have emerged in China, leading to job diversification. Currently, the overall development status of new occupations in China and the structural characteristics of new occupation practitioners in different cities are still unclear. This study first constructed a development index system for new occupation practitioners from five dimensions (group size, cultural appreciation, salary level, occupation perception, and environmental perception). Relevant data to compare and analyze the development status of new occupation practitioners were derived from the big data mining of China's mainstream recruitment platforms and the questionnaire survey of new professional practitioners which from four first-tier cities and 15 new first-tier cities in China. The results show that the development level of new occupation practitioners in the four first-tier cities is the highest, and the two new first-tier cities, Chengdu and Hangzhou, have outstanding performance. The cities with the best development level of new occupation practitioners in Eastern, Central, and Western China are Shanghai, Wuhan, and Chengdu, respectively. Most new occupation practitioners in China are confident about the future of their careers. However, more than half of the 19 cities are uncoordinated in the five dimensions of the development of new occupation practitioners, especially those cities with middle development levels. A good policy environment and social environment have not yet been formulated to ensure the sustainable development of new occupation practitioners. Finally, we proposed the following countermeasures and suggestions: (1) Establish a classified database of new occupation talents. (2) Implement a talent industry agglomeration strategy. (3) Pay attention to the coordinated development of new occupation practitioners in cities.

## Introduction

The Ministry of Labor and Social Security of China defines new occupation as an occupation already existing in economic and social development with a certain scale of employees and relatively independent vocational skills, but has not been included in the "Classification of Occupations of the People's Republic of China". The employment situation of China had been

Department (2022JDR0024). The funder
supported writing the paper.

**Competing interests:** The authors have declared
that no competing interests exist.

affected by COVID-19 pandemic. According to the data of Ministry of Human Resources and Social Security, Chinese new employee population decreased year-on-year by 950,000 in the first quarter. Employment is the basic implications for national development, a challenging employment environment has resulted in millions of individuals losing their jobs, flexible boundary new occupations are proliferating fast. The digital economy presently be a powerful driving force for economic progress in the period of kinetic energy transformation of social economic [1–4]. Researchers, business leaders, and governments have recognized the tremendous potential of the digital economy. And new terms such as Industry 4.0 [5], Productivity 4.0, and Society 5.0 [6] have emerged and been accepted by the public. China's "14th Five-Year Plan" pointed out that we must accelerate the digitization process, develop a digital economy, promote digital industrialization and industrial digitization, promote the deep integration between the digital economy and the real economy, and create an internationally competitive digital industry cluster [7].

Faced with the emergence and rapid dissemination of an internet-based digital economy, new occupation practitioners have emerged, and millions of workers have involved in this new employment field. According to the White Paper on Development and Employment of China Digital Economy [8], people who worked in the digital economy amounted to 191 million in 2018, accounting for 24.6% of total employment that year [8]. Meanwhile, employment forms and patterns have changed, and flexible working hours are arranged to complement the performance of other work, like family-related activities [9]. Employees transit from traditional occupations to new occupations. The "new occupation" context is depicted as one where transitions occur more frequently than ever, leading to discontinuities and fragmented occupation [10]. Based on Human-Environment Fit, the current situation and future development trend of new occupation practitioners under the background of the digital economy are discussed by this paper. Traditional economic development was slowing down. The new economy, represented by the new generation of technologies such as mobile internet, cloud computing, big data analytics, and artificial intelligence (AI), has emerged [11].

New occupation practitioners are the driving force of the data economy. Therefore, qualitative and quantitative research on new occupation practitioners is crucial. This study constructed a development index system for new occupation practitioners based on five dimensions (group size, cultural appreciation, salary level, occupation perception, and environmental perception). This paper aims to explore the current situation and existing problems of new occupation practitioners in major cities of China by comparing the development level of new occupation practitioners in major Chinese cities.

This study makes two major contributions. First, a new occupation practitioner index system construct to measure the development level of new occupation practitioners. Second, existing problems are identified through comparative analysis. Furthermore, put forward to promote the sustainable development of new occupation practitioners' countermeasures and suggestions.

The rest of this paper organized as follows. Section 2 discusses the relevant literature on the new economy and new occupations. Section 3 describes the index system and model construction and explains the research object and data sources. Section 4 presents the calculation and analysis result analysis. Section 5 presents conclusions and policy implications.

## Literature review

### New economy

The term "new economy" first appeared in Business Week, which refers to the information technology revolution and the economic form driven by it, with high-tech industries as the

leading role under the background of economic globalization [12]. The new economy, also called the digital economy, represents the economic and technological development frontier [13] and is characterized by high economic growth, economic structure optimization, and upgrading [14]. Its operational law differs from that of the traditional economy, especially in terms of foundation support, technical characteristics, organizational structure, and industrial organization [15, 16]. Poutanen and Kovalainen believed that the core connotation of the "new economy" was innovation-driven [17]. Innovation is the core driving force for the generation and development of economic activities and could promote the transformation and upgrading of economic structures. To date, the development of the new economy has become a new source for major global powers to shape their unique international competitiveness [18]. Currently, the digital economy pervades insurmountable prospects for the world economy, influencing different sectors such as energy, banking, retail, publishing, transportation, education, health, media, culture, and e-government development [19]. Gregory studied the reform of for-profit university education in this era of the new economy [20]. The integration of the internet and the real economy has propelled the formation of a new economy. Internet cultural industries, defined as a collection of industries that engage in creative cultural content, production, circulation, and services based on internet technology and core digitization, are a critical part of this new economy [21]. Digital revolutions in medicine are part of the complex digital economy based on creating value from the analysis of behavioral data acquired by tracking daily digital activities, which offers exciting new directions for the treatment of mental illness [22]. Other scholars have also investigated the new economy and new driving forces from the perspective of statistical research and pursued the high-quality development of the Chinese economy [23]. Wang et al. [24] examined the impact of the new economy on the distribution of urban sizes in 102countries. The results show that the new economy has a significant impact on the distribution of urban size, and this impact is different in different countries. Nham and Ha research results show that the initial development of digitization can enable European countries to transition to a circular economy. However, the excessive development of digitization hinders this process [25]. Digital economy is of great significance to carbon emission reduction [26]. There is an inverted U-shaped relationship between digital economy and carbon emissions. Similarly, the spatial spillover effect of digital economy on carbon emissions is also inverted U-shaped [27]. At this time, the Chinese economy has transformed from an era of the consumer internet to the industrial internet, with data becoming the core production element [28]. Using big data analytics capabilities could provide real-world evidence of the potential existing challenges in the digital economy [29]. And some scholars have also studied the relationship between the incidence of digital economy and long-term frictional unemployment in various countries. The results show that there is a strong negative partial correlation between national unemployment rate and the incidence of digital economy [30].

## New occupation

Initially, the United States had the best new career development because its culture emphasized independence and self-realization, and its labor market practices were quite flexible and open [31, 32]. Odd jobs account for an increasing proportion of the labor force, and researchers are more and more interested in it [33]. However, today's society is becoming increasingly "mobile" and uncertain [34]. Other countries are also interested in this [35]. The fourth scientific and technological revolution, with the rise of emerging fields such as the internet plus, big data, and AI, has brought about economic transformation and industrial restructuring. Moreover, the in-depth reform of the economic structure has triggered systematic changes in employment, while industrial transformation and upgrading have spawned large amounts of

new jobs. The Ministry of Human Resources and Social Security, the State Administration for Market Regulation, and the National Bureau of Statistics jointly released 13 new occupations on April 1, 2019. Another 25 new occupations were released in 2020. The fourth technological revolution, with internet IT at its core, is the immediate inducer of the birth of new occupations [36]. With the advent of the internet, big data, and AI eras, a large number of occupations requiring technical skills are gradually being replaced by intelligent robots. However, the increase in the number of workers with high and low technical skills shows job polarization [37]. The new occupation forms around the need to develop AI technology and people's leisure services [38]. Liu proposed that the rise of new occupations covered social and economic reasons [39]. On the one hand, with the rise of IT, society has entered a new stage of the combination of virtual economy and the real economy, resulting in a number of new occupations arising from the transformation of social structure. On the other hand, the development of IT has driven the industrial upgrading of the "Internet and industry," with the emergence of new jobs such as couriers, network anchors, smart tour guides, and personal caregivers. Rivas-Gayo found that electronic health records were introduced in almost all healthcare settings to end traditional paper records, leading to the emergence of the new profession of health document managers [40]. Using the case study method, Patrikakis & Murugesan explored the extensive impact of the digital economy on occupations and the skills required for new occupations [41]. However, some scholars have also discussed the influence of employees' attitudes toward new occupations on subjective occupation success. For instance, Kundi et al., based on career motivation theory and job crafting theory, demonstrated the differentiated effect of employees' new occupation attitudes on subjective occupation success in terms of occupation commitment and occupation satisfaction, focusing on two new occupation attitudes, namely protean occupation, and boundaryless occupation [42]. These new occupation practitioners can get very flexible and potentially autonomous jobs, but they also have to deal with the challenges posed by the nature of the job, its instability and their relationship with the platform business [43]. Jabagi et al. [44] drew lessons from self-determination theory, job characteristic theory and enterprise social media research, this paper explores how the architecture of digital labor platform behind digital economy affects the key factors of self-motivation. Smith et al. [45] discussed Australian consumers' attitude and understanding of working conditions and rights in APP-based delivery services, when a large proportion of people are willing to pay more to improve the income and working conditions of takeout workers, but at the same time it finds that the money they are willing to pay is unlikely to lead to continuous improvement in working conditions. Zipperer et al. [46] analyzed the influence of contractors registered on Uber and other digital platforms on social security, fair wages and job stability of practitioners, and puts forward public policy suggestions to improve the working conditions of new occupation practitioners.

These people engaged in new occupations are collectively referred to as new occupation practitioners. Mousa and Chaouali [47] through a survey of new occupation practitioners registered on three crowdsourced platforms, the results show that active behavior in personal and collaborative workflows may make new occupation practitioners feel meaningful, which in turn makes them emotionally committed to the crowdsourced platform they registered. Keith et al. [48] through the survey of new occupation practitioners, those who use new occupation as the main source of income have lower income, longer time to complete a lot of work, and lower level of satisfaction with current and future provincial capitals. As a group engaged in a new occupation, there is currently little research in the academic circle. While the available literature has discussed the origin, development trend, and classification of the new occupation from the theoretical aspect, the research on new occupation practitioners still needs to be further explored. By constructing a development system for new occupation practitioners in

China and based on the existing data, this study comprehensively evaluates the development level of new occupation practitioners in the main cities of China. To better evaluate the development status of the new occupation practitioners in China and discover relevant issues during the development. Such a survey is important because it is an important source of job satisfaction [49].

## Method

### Index system and model construction

The analytic hierarchy process is a systematic and hierarchical analysis method that combines qualitative and quantitative analyses [50]. The characteristic of this method is that based on an in-depth study of the nature, influencing factors, and internal relations of complex decision-making problems, the thinking process of decision-making is mathematized using less quantitative information. Thus, it provides a simple decision-making method for complex decision-making problems with multi-objectives, multi-criteria, or no structural characteristics. The specific steps are as follows:

**Construction of index system.** The evaluation index system refers to an organic system with an internal structure that is composed of a number of indicators that characterize the characteristics of the evaluation object and its interrelations. It is widely used in various fields such as green mining [51], water problems [52], human competence [53], and so on. By sorting out the relevant literature, Zhao [54] deemed that an industry's professionalism comprised five factors: systematic knowledge system, professional moral creed, judgment standard, culture, and social recognition. Professionalism is supported by a set of professional values, attitudes, and behaviors. Tong [55] proposed that the evaluation criteria for China's new-type professional farmers should include five indicators: professional cultural accomplishment, salary, skills, behavior norms, and recognition. Based on the characteristics of the new occupation group, we determined five first-level indicators in the evaluation index system: group size, cultural appreciation, salary level, occupation perception, and environmental perception.

The most basic premise for the formation of an occupation is that the profession reaches a certain scale. Therefore, occupation development is inseparable from the development of occupation size. Professional cultural appreciation refers mainly to the level of cultural knowledge of practitioners, which is the basic quality for practitioners to adapt and develop in their occupations [56]. Salary is generally divided into skill-based compensation [57], position-based compensation, and competency-based compensation. Professional salary is one of the main factors influencing occupation differentiation and the social division of labor—occupation perception includes recognition and satisfaction. The recognition and satisfaction of a profession are closely related to its social status, working environment [58], work intensity, income level, vocational welfare, development space, professional characteristics, etc. Environmental perception is also very important in the development of new occupations. Occupation development is inseparable from the support of the government, regional culture, climate, and so on. Based on the characteristics of new occupations, this study puts forward the development indicators of new occupation practitioners, including five first-level indicators of new occupation practitioners' group size, literacy, occupation salary, occupation perception, and environmental perception. The second-level indicators were determined through expert interviews and data; the specific indicators are shown in Fig 1.

**Data processing and model construction.** In order to eliminate the impact of diverse dimensions on the results, the range method was employed, and the results were controlled

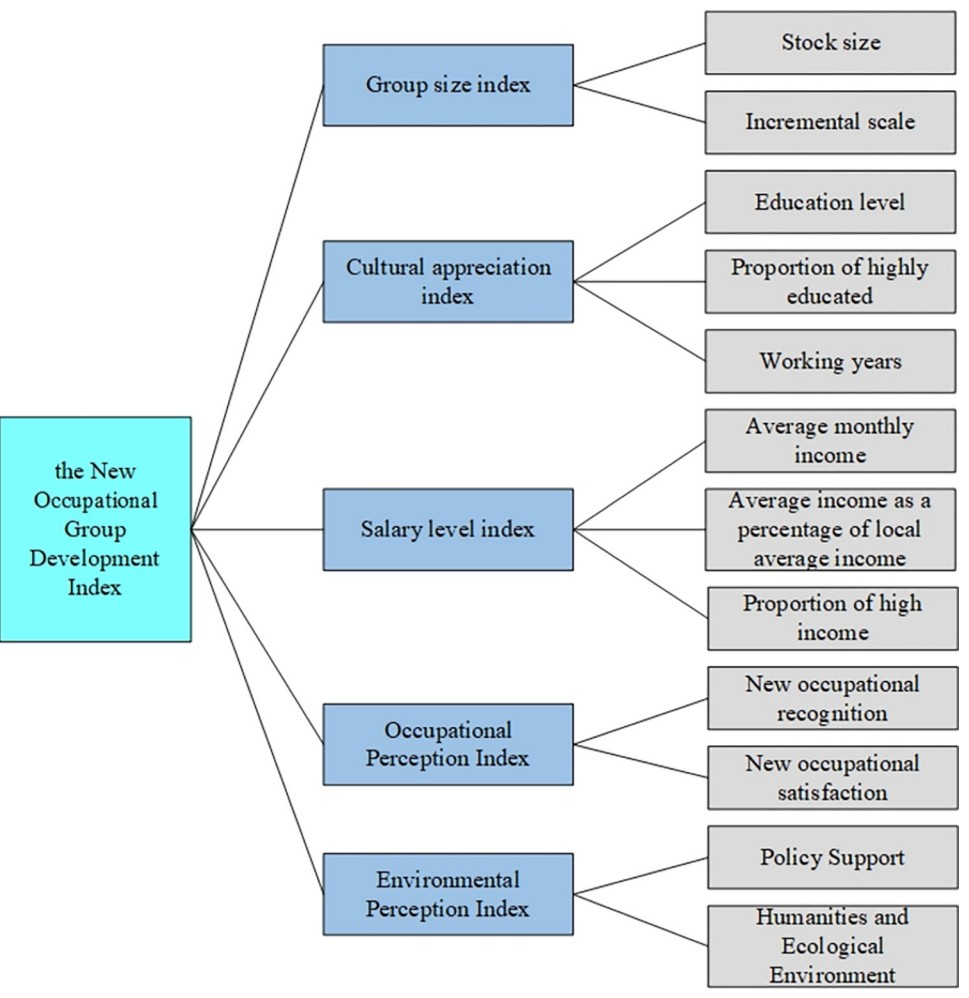

**Fig 1. Index system of new occupation practitioners.**

within [60,100] [59]. The formula is as follows:

$$\bar{e}_{ij} = \frac{e_{ij} - \min(e_{ij})}{\max(e_{ij}) - \min(e_{ij})} \times 40 + 60, \tag{a}$$

In Eq ($a$), $e_{ij}$ and $\bar{e}_{ij}$ represent the original data and dimensionless data of city j in the $i$ year, respectively.

In the multi-index comprehensive evaluation, it is necessary to synthesize each index through a mathematical model, which generally employs a mixed composite model combined with a weighted geometric average composite model, a weighted arithmetic average composite model, and a mixed composite model. Considering the characteristics of the index and the convenience of operation, a weighted arithmetic mean model was applied to synthesize the index [60]. The calculation formula of first-level indicators is as follows:

$$Index_{Bi} = \sum_{i=1}^{j} I_{Ci} \times W_i, \tag{b}$$

**Table 1. Introduction of 19 "first-tier" cities.**

| Sort | City |
|---|---|
| First-tier cities | Shanghai, Beijing, Guangzhou, Shenzhen |
| New first-tier cities | Chengdu, Hangzhou, Nanjing, Wuhan, Suzhou, Zhengzhou, Xi'an, Chongqing, Tianjin, Hefei, Dongguan, Changsha, Shenyang, Foshan, Qingdao |

In Eq (b), $Index_{Bi}$ represents the $Bi$ Index in the first-level Index, $I_{Ci}$ indicates the second-level Index, $W_i$ refers to the weight of the corresponding second-level Index, and j is the number of indicators.

The index system of this study is composed of two layers, and the bottom-up method was adopted to obtain the development index of new occupation practitioners in Chinese cities. The calculation formula is as follows.

$$Index_{NOP} = \sum_{i=1}^{j} Index_{Bi} \times W_i, \tag{c}$$

In Eq (c), $Index_{NOP}$ means the development index of new occupation practitioners, $W_i$ indicates the weight of the first-level city index, and j represents the number of indicators.

## Study subject and data source

**Study subject.** The first-tier city refers to the metropolis, which plays an important role and a leading role in national political, economic, and other social activities. This study took new occupation practitioners in 19 "first-tier" cities in China as the research object. Table 1 lists these 19 first-tier cities. China's Ministry of Human Resources and Social Security released 38 new occupations in 2019 and 2020, which were selected, respectively. Table 2 presents the results. These new occupations involve fields that are wide. They are mainly engineers and technicians engaged in emerging industries, such as big data, AI, the Internet of Things (IoT), cloud computing, and blockchain. There are also new occupations related to healthcare, such as healthcare nurses and respiratory therapists. New occupations related to the service industry, such as online delivery service provider and online learning service provider, were also included.

**Table 2. New occupations for 2019 and 2020.**

| Year published | New occupation |
|---|---|
| 2019 | AI engineers and technicians, IoT engineers and technicians, big data engineers and technicians, cloud computing engineers and technicians, digital managers, building information model technicians, e-sports operators, e-sports players, drone pilots, agricultural managers, IoT installation and debugging personnel, industrial robot system operators, industrial robot system operation and maintenance personnel |
| 2020 | Intelligent manufacturing engineering and technical personnel, industrial Internet engineering and technical personnel, engineering and technical personnel of virtual reality, chain operation management, supply chain management division, network Distributor, AI trainers, electrical and electronic products environmental inspector, all media operations division, health according to nurses, respiratory therapists, congenital disability prevention and control, rehabilitation assistive technology consultants, consultants, installation of unmanned aerial vehicle, high-speed railway line comprehensive maintenance workers, prefabricated building construction workers, blockchain engineers and technicians, blockchain application operators, urban management grid workers, Internet marketers, information security testers, online learning service engineers, community health assistants, older adults ability assessors, additive manufacturing equipment operators |

**Data source.**

1. Big data mining. Liepin and Ganji are well-known job recruitment platforms in China. They hold data on demand for new occupations in the market. Using big data mining of the whole network, such as Liepin (https://www.liepin.com/) and Ganji (https://www.ganji.com/), etc., we obtained the annual demand information of new occupation practitioners in 19 "first-tier" cities since the emergence of a large number of new jobs and obtained a total of 7.14 million data in 2020.

2. Questionnaire survey. Through the questionnaire survey method, we gathered information related to the living and working status of new occupation practitioners in China, which provides data support for the eigenvalue portrait and the analysis of the living status and the change in job perception of new occupation practitioners. The survey objects of the questionnaire were the people engaged in the 38 new occupations, while the content is the basic information of the new occupation practitioners, including gender age, education, occupation, work experience, etc., as well as the perception of relevant policies and the degree of satisfaction and recognition of the new occupation practitioners. Eventually, 5,106 questionnaires were obtained from 19 cities, of which 4,273 were valid, with an effective recovery rate of 83.69%. The valid questionnaire revealed that 65.75% were male participants and 34.25% were female participants. As for age, people aged between 18 and 24 took up 32.60%, between 25 and 29 were 31.72%, and 92.87% were under the age of 40.

## Empirical analyses

### Index weight calculation

In this study, the analytic hierarchy process was used to calculate the weight of each index. Twenty experts and scholars in fields related to the new economy and new occupations were invited to score using the 1–9 scale method. This study employed Python software for calculation and consistency check of the judgment matrix to ensure CR < 0.1. The weights of each index in the development index system of new occupation practitioners in China are presented in Table 3.

### Calculation and analysis of various indicators

According to the relevant data of the second-level indicators, Eq (a) was applied to the process, and Eq (b) was used to calculate the sub-index of the first-level indicators. Finally, Eq (c) was

**Table 3. The weight system of development indicators of new occupation practitioners in China.**

| First-level index | Weight | Second-level index | Weight | Comprehensive weight |
|---|---|---|---|---|
| Group size index(B1) | 0.3228 | (C1) Stock size | 0.6667 | 0.2152 |
|  |  | (C2) Incremental scale | 0.3333 | 0.1076 |
| Cultural appreciation index(B2) | 0.1267 | (C3) Education level | 0.1958 | 0.0248 |
|  |  | (C4) Proportion of highly educated | 0.4934 | 0.0625 |
|  |  | (C5) Work experience | 0.3108 | 0.0394 |
| Salary level index(B3) | 0.1469 | (C6) Average monthly income | 0.4934 | 0.0725 |
|  |  | (C7) Average income as a percentage of local average income | 0.3108 | 0.0457 |
|  |  | (C8) Proportion of high income | 0.1958 | 0.0288 |
| Occupation perception index(B4) | 0.1897 | (C9) New occupation recognition | 0.6667 | 0.1265 |
|  |  | (C10) New occupation satisfaction | 0.3333 | 0.0632 |
| Environmental perception index(B5) | 0.2140 | (C11) Policy Support | 0.6667 | 0.1427 |
|  |  | (C12) Humanities and Ecological Environment | 0.3333 | 0.0713 |

**Table 4. Development Indicators of the new occupation practitioners in China.**

| City | B1 | B2 | B3 | B4 | B5 | $Index_{NOP}$ |
|---|---|---|---|---|---|---|
| Shanghai | 94.86 | 96.14 | 87.92 | 96.80 | 94.14 | 94.23 |
| Beijing | 95.97 | 90.60 | 89.06 | 100.00 | 80.44 | 91.72 |
| Shenzhen | 84.16 | 81.15 | 83.55 | 93.32 | 99.13 | 88.64 |
| Guangzhou | 91.14 | 79.10 | 78.89 | 86.66 | 82.18 | 85.06 |
| Chengdu | 86.39 | 77.49 | 70.96 | 78.73 | 89.74 | 82.27 |
| Hangzhou | 77.89 | 75.85 | 90.91 | 90.32 | 72.18 | 80.69 |
| Nanjing | 70.22 | 76.96 | 77.53 | 81.99 | 69.81 | 74.30 |
| Wuhan | 74.16 | 74.92 | 76.63 | 80.31 | 65.93 | 74.03 |
| Suzhou | 66.93 | 73.41 | 71.44 | 83.01 | 71.17 | 72.38 |
| Zhengzhou | 66.30 | 70.16 | 67.75 | 87.07 | 66.25 | 70.94 |
| Xi'an | 70.19 | 72.92 | 74.39 | 77.80 | 61.67 | 70.78 |
| Chongqing | 64.14 | 65.83 | 70.08 | 84.71 | 64.94 | 69.31 |
| Tianjin | 61.73 | 66.02 | 76.02 | 88.47 | 60.36 | 69.16 |
| Hefei | 63.19 | 74.15 | 83.39 | 65.13 | 62.66 | 67.81 |
| Dongguan | 65.39 | 63.42 | 70.21 | 76.68 | 62.16 | 67.31 |
| Changsha | 63.86 | 65.73 | 72.11 | 73.41 | 64.08 | 67.17 |
| Shenyang | 60.37 | 67.70 | 67.85 | 60.37 | 70.49 | 64.57 |
| Foshan | 60.55 | 61.44 | 69.06 | 67.48 | 61.66 | 63.47 |
| Qingdao | 62.32 | 69.26 | 65.70 | 60.00 | 60.76 | 62.93 |

employed to calculate the comprehensive index. This represents the development index of China's new occupation practitioners in 2020. SPSS was used to analyze the data.

The composite index shows that in 2020, there was a large gap in the development of new occupations among cities in China in Table 4. The development of new occupation practitioners in the four first-tier cities is ahead of that in the 15 new first-tier cities. Among the 15 new first-tier cities, only Chengdu and Hangzhou had a new occupation practitioner development index above 80. They opened up a distance from other first-tier new towns. However, in the eight cities, the development index of new occupation practitioners is below 70. According to the comprehensive index, the development level of new occupation practitioners in 19 cities was divided into three levels: high, medium, and low. Table 5 lists the specific divisions. The development level of new occupation practitioners in the eastern region was higher than that in the Central and Western regions. In the Eastern region, Shanghai has the highest development level of new occupation practitioners, reaching 94.23. In the Central region, Wuhan has the highest development level of new occupation practitioners, reaching 74.03. In the Western region, Chengdu has the highest development level of new occupation practitioners, reaching 82.27.

The five sub-indexes show that, among the 19 cities, Beijing has the highest group size index and occupation perception index. Shanghai has the highest cultural literacy index. Hangzhou has the highest salary index. Shenzhen has the highest environmental perception index. The Shenyang group has the lowest group size index. Foshan has the lowest cultural literacy index. The city with the lowest salary index, occupation perception index, and environmental perception index is Qingdao. The group size index shows a large gap between the 19 cities, and the group size index of the 10 cities is less than 70. Notably, among the five dimensions of the development of new occupation practitioners in each city, 11 cities have the highest occupation perception index. New occupation practitioners are more satisfied with their new occupations and are more optimistic about the future of their new occupations. Nine cities have the lowest environmental perception index, which shows that a good environment has not yet been formed to promote the sustainable development of new occupation practitioners.

Table 5. The development level of new occupation practitioners in China's "first-tier" cities.

| Development level | City | Index interval |
|---|---|---|
| High | Shanghai, Beijing, Shenzhen, Guangzhou, Chengdu, Hangzhou | 80–100 |
| Medium | Nanjing, Wuhan, Suzhou, Zhengzhou, Xi'an | 70–80 |
| Low | Chongqing, Tianjin, Hefei, Dongguan, Changsha, Shenyang, Foshan, Qingdao | 60–70 |

Fig 2 clearly shows that the occupation perception index of new occupation practitioners in most cities is high; however, the environmental perception index is low, especially in cities where the development of new occupation practitioners is at a medium level. In cities where the development of new occupation practitioners is at a high or low level, there is little fluctuation in each dimension. In cities where the development of new occupation practitioners is at a medium level, all dimensions fluctuate significantly.

## Analysis of the coordination of indicators of various cities

New occupation practitioners in Chinese cities have uncoordinated development on the five indices of group size, cultural appreciation, salary level, occupation perception, and environmental perception. In this study, radar charts were created according to each component (as shown in Fig 3) to analyze the coordination level of the crowd of the cities' new occupation development.

Among the four first-tier cities, Shanghai has the best coordination degree in the five indicators development. Beijing and Guangzhou are not far behind, group size, cultural appreciation, salary level, occupation perception all developed better, and the environmental perception index was a common problem. The development of environmental perception and occupation perception in Shenzhen were good, while the size, cultural appreciation, and salary level perform slowly. Chengdu, a new first-tier city with the best development of new

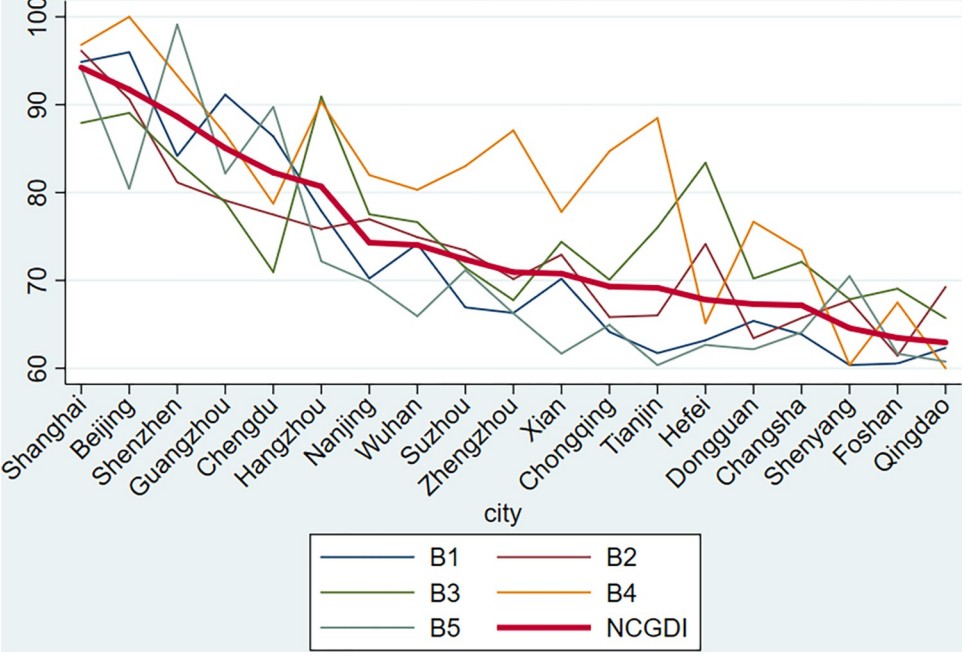

Fig 2. Indicators of the development of new occupation practitioners in China.

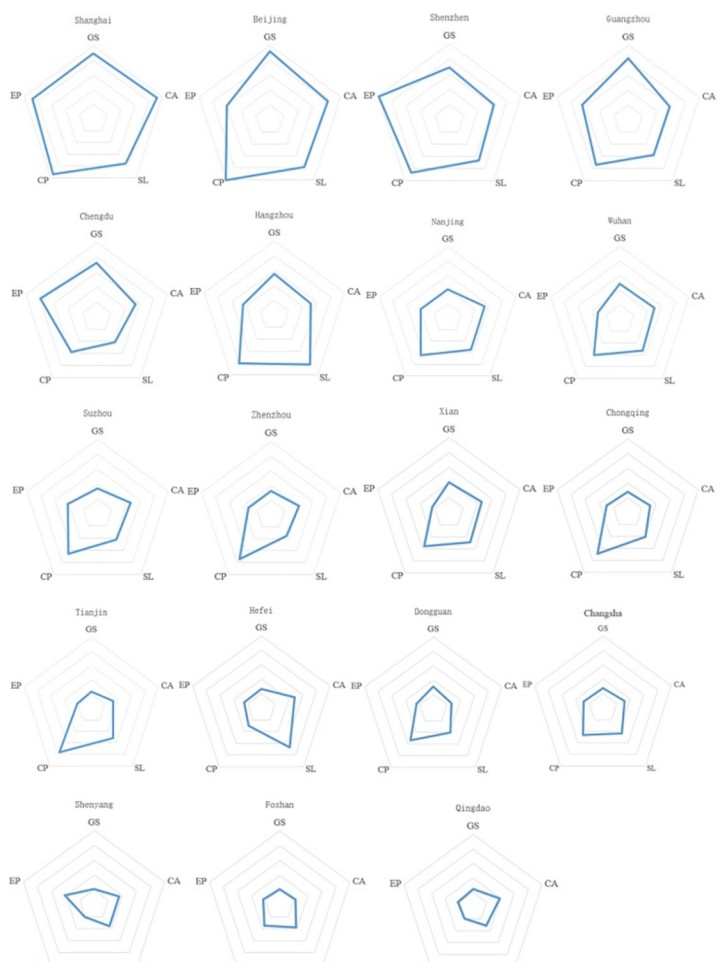

**Fig 3. Radar chart of coordination degree of five indicators in Chinese cities.**

occupation practitioners among the 15 new first-tier cities, has a high degree of development coordination, only salary level developed slowly. The Southeast coastal cities of Hangzhou, Nanjing, Suzhou, Dongguan, and Foshan developed well in the three indexes of cultural appreciation, salary level, and occupation perception. In contrast, group size and environmental perception both are in a weakened state. Wuhan and Xian City developed well in terms of group size, cultural appreciation, salary level, and occupation perception, with poor development of environmental perception. The coordination degree of five indexes of Zhengzhou, Chongqing, Tianjin, and Hefei was low, occupation perception index with Zhengzhou, Chongqing, and Tianjin was the only one higher index. In contrast, Hefei had a better development in salary level. In Changsha, Shenyang, and Qingdao, the development of new occupation practitioners was relatively coordinated in the five indicators, but it came late and developed slowly.

## Discussion

In this study, most new occupations are borderless. Their work is very flexible. For example, online delivery staff and ride-hailing drivers can work according to their own time. There are fewer restrictions on working hours and workplaces. They are more likely to have multiple jobs, at the same time, they tend to be younger [61]; this is consistent with our research results.

New occupation practitioners are more satisfied with their careers, and Kundi [42] showed that this kind of career practitioner has higher satisfaction.

The results show that new occupation practitioners in the four first-tier cities have a higher level of development. Cities with better economic development are more conducive to developing new occupations. Cities with better economic development can provide a more favorable environment for the development of the new occupation. At early time, some scholars proposed that borderless would become the main career model soon [62], which has become a reality. In recent years, its scale has expanded. An increasing number of people benefit from such careers [63].

## Conclusion and policy implications

To better evaluate the development status of new occupation practitioners in China, discover relevant issues during development, and provide a reference for the development of a digital economy, this study measured the development level of new occupation practitioners in 19 "first-tier" cities in China from five dimensions: group size, cultural literacy, salary level, occupation perception, and environmental perception. Based on a comparative analysis of 19 "first-tier" cities, this study describes the development level of new occupation practitioners in China. The results indicate that the new occupation practitioners have a high degree of recognition of and satisfaction with the new occupations and hold a positive and optimistic attitude toward the development prospects of the new occupation practitioners. The four first-tier cities lead 15 new first-tier cities in the development level of new occupation practitioners. Among the 15 new first-tier cities, Chengdu and Hangzhou were the best in the development of new occupation practitioners. Comparatively, other new first-tier cities had a relatively slow start and late development of new occupations. In most cities, the development of new occupation practitioners was not coordinated with the five indicators, severely affecting the healthy development of new occupation practitioners. The policy strength of relevant departments in some cities was low and has not taken the shape of a good policy environment and social environment to promote the sustainable development of new occupation practitioners. To promote the healthy development of new professional practitioners. Therefore, this study proposes the following countermeasures and suggestions.

1. Analyze and sort out the changes in demand for new occupation practitioners from technological progress, industrial upgrading, and economic and social development. Attention should be paid to the cultivation of new occupation practitioners and accurately breeding related research and development (R&D) personnel to achieve precise assistance based on the layout of the new economy industry. Surrounding relevant industries related to national core competitiveness, such as 5G, new infrastructure, big data, intelligent manufacturing, and AI, with the new economic industry chain and project sources to attract talent, "integration between industries and talents" should be further promoted create talent cluster, form talent chain, set up to attract, to promote the production, to produce to merge the benign pattern, around "industrial chain" to make the chain of "talent" talent'.

2. Establish a classified database for new occupation talents. According to the hierarchy and contribution of new occupation practitioners, we can provide a guarantee of classification grading priority service in the household registration, housing security, children education, investment, and health care to form a good situation with an "environment to attract talent, service to retain talent," open up a good occupation development environment for R&D staff and provide the broad space for development, build first-class development platform.

3. Implementing a talent industry agglomeration strategy. Regional talent competitiveness can be converted into regional industrial competitiveness and regional economic competitiveness by focusing on strengthening talent science and technology platforms. All regions take the industry-university-research platform, attract talent, achieve technological innovation, incubate innovative enterprises, and nurture innovative and entrepreneurial talent. Additionally, there is a need to optimize the mechanism for the flow of talented people, further remove the institutional obstacles to the flow of talented people, let there be a free flow of talented people truly, and activate the new economy and new occupation development of the country.

4. Paying attention to the coordinated development of new occupation practitioners in cities, we suggest carrying out classified policies based on the different development stages of new occupation practitioners in various cities. With the rapid development of cities, we should guide the high-quality development of new occupation practitioners, offer occupation practitioners, and provide more policy support in terms of value realization. For relatively backward cities, it is necessary to provide a more tolerant and convenient entry threshold for new occupation practitioners, introduce more innovative and entrepreneurial support policies related to the new economy and new occupation practitioners, and strengthen the introduction and cultivation of talents for new occupations as the main starting points, to expand the space for new occupation development in cities and open up new occupation development channels. At the same time, it is also necessary to strengthen the publicity and promotion of the new occupation, improve social awareness of the new occupation, and enhance the social image of the new occupation.

The digital economy not only impact on China but has a profound impact on the world-wide. The development of new professional practitioners relates to the development of the digital economy. Every Country should be aware of the current situation that the development of new professional practitioners. Regulate and control the healthy development of the digital economy.

## Supporting information

**S1 Data.**
(XLSX)

## Author Contributions

**Conceptualization:** Yuxiang Zhang.

**Data curation:** Anhang Chen.

**Funding acquisition:** Huiqin Zhang.

**Investigation:** Linzhen Li.

**Methodology:** Anhang Chen.

**Project administration:** Yuxiang Zhang.

**Supervision:** Huiqin Zhang.

**Writing – original draft:** Linzhen Li.

**Writing – review & editing:** Yuxiang Zhang, Huiqin Zhang.

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
