## [Decision Letter · Decision Letter 0]

26 May 2022

PONE-D-22-01167Comparative study on the development of new occupation practitioners in China's “first-tier” citiesPLOS ONE

Dear Dr. Zhang,

Thank you for submitting your manuscript to PLOS ONE. After careful consideration, we feel that it has merit but does not fully meet PLOS ONE’s publication criteria as it currently stands. Therefore, we invite you to submit a revised version of the manuscript that addresses the points raised during the review process.

We look forward to receiving your revised manuscript.

Kind regards,

Ziqiang Zeng, Ph.D.

Academic Editor

PLOS ONE

Journal Requirements:

3. Please remove your figures from within your manuscript file, leaving only the individual TIFF/EPS image files, uploaded separately.  These will be automatically included in the reviewers’ PDF.

Reviewers' comments:

Reviewer's Responses to Questions

**Comments to the Author**

1. Is the manuscript technically sound, and do the data support the conclusions?

Reviewer #1: No

Reviewer #2: Yes

2. Has the statistical analysis been performed appropriately and rigorously? 

Reviewer #1: Yes

Reviewer #2: I Don't Know

3. Have the authors made all data underlying the findings in their manuscript fully available?

Reviewer #1: Yes

Reviewer #2: Yes

4. Is the manuscript presented in an intelligible fashion and written in standard English?

Reviewer #1: Yes

Reviewer #2: No

5. Review Comments to the Author

Reviewer #1: 1. Abstract: Can the background and limitations of previous studies in the original text be more specific? The author can explain this part in more detail.

2. Abstract: The abstract of the original text only explains the method of data acquisition, but does not explain the source of data. The author should supplement this part.

3. Abstract: The enlightenment and significance in the abstract should be more clear and have specific significance.

4. The author can add the word "questionnaire survey" to the keyword.

5. Introduction: In the original text “Chinese "14th Five-Year Plan" points out that we have to accelerate the digitization process, develop digital economy, promote a digital industrialization and industrial digitization, promote the deep integration between the digital economy and the real economy, and create an internationally competitive digital industry cluster.” Relevant documents of the policy should be supplemented here and added to the references。

6. Introduction: The theoretical basis and research content of this study are not described in detail in the original text. The author should make a detailed supplement.

7. Introduction: The length of this part is large, and there is no certain correlation between the previous and subsequent paragraphs, which will give people the feeling of unclear logic and disorder. It is suggested that the author sort out the introduction by establishing a secondary title.

8. Literature review: In the literature review section, the author should write around the scientific problems mentioned above, so that this part has a clear logical line.

9. Literature review: The author can summarize the literature review part by list.

10. Method: The part of "index system construction" in the original text should be one of the important parts of this paper. The references cited in this part are documents of earlier periods. The author can supplement the relevant authoritative documents in recent years。

11. Method: In the "data processing and model construction" part of the original text, the author should supplement the references of using this data processing method, and please explain the reasons and advantages of using this data processing method in combination with the content of this article。

12. Method: In the "data source" part of the original text, the author should supplement the relevant websites and database links of the data source. At the same time, the author should supplement the questionnaire and explain the reliability and validity of the questionnaire.

13. Method: Empirical analyses

14. Method: The author should explain the software used in the empirical study.

15. Method: The author should place Table 4 on the same page.

16. The author can supplement a Discussion section to compare the results of this paper with those of predecessors.

17. Conclusions: The research purpose, content and methods of the article should be described in the conclusion of the original text.

18. Conclusions: When expressing the research results, the language can be appropriately simplified. At the same time, the enlightenment given by the author should go beyond the Chinese perspective.

Reviewer #2: This is an interesting manuscript that examines the emergence of a number of new professions in China and how they have developed in 19 first tier cities. I have a number of suggestions that will hopefully improve the manuscript.

In the first line of the abstract, you mention the complex economic environment and severe employment situation, but this doesn't really seem to come through in the body of your work.

I found the introduction and literature review to lack coherence - I wasn't always sure of what the 'story' you are trying to tell is. One suggestion is to use sub-headings to help organise your thinking and writing in a logical way. At the end of the introduction you should state your clear aims/objectives/research questions/hypotheses for this work.

What are 'first tier cities'? an international audience wouldn't be likely to understand what they are without some explanation.

It might be worth introducing things in a reverse chronological order - what started the "new economy" and how has it progressed to the digital economy and beyond? What are the key features of each of these 'phases'?

Line 128 - five new what?

An example at line 129 - 'economy is developing in full swing' - this is very jargon/colloquial and should be re-written. The audience in PLOS One is broad and won't necessarily know or understand the jargon. There are instances of this throughout the manuscript.

Section 3.1.1 - I would suggest starting with a brief explanation of what the index system is, why it is needed/important and how it works (at a high level). Then get into the details of your index.

Lines 208-227 - this is some good background/context to the index, however, there are no references. Please reference this section appropriately.

At line 271, what are the Liepin and Ganji networks? What data do they hold? How were they mined? What are they?

Lines 286-288 - report the demographics in a consistent way - either by age or by a birth year range, not a mix of the two.

Overall, the manuscript requires a thorough English language proofing; this will make it easier to assess its academic merits.

6. PLOS authors have the option to publish the peer review history of their article (what does this mean?). If published, this will include your full peer review and any attached files.

Reviewer #1: No

Reviewer #2: No

---

## [Author Response · Author response to Decision Letter 0]

24 Jul 2022

Dear editors and reviewers,

We would like to thank you and the two reviewers and editors for their valuable comments and kind suggestions regarding the manuscript we submitted. These comments and suggestions directed us to improve the content of our study. We have revised our paper substantially, and the revised version of the manuscript incorporates the comments and suggestions of the reviewers. 

The details of modification were listed point by point:

Response to Reviewer #1:

[Comment 1] Abstract: Can the background and limitations of previous studies in the original text be more specific? The author can explain this part in more detail.

[Response] Thank you very much for your comments. We have added the relevant content according to your opinion, as shown below.

" Under the background of increasingly complex economic environment and severe employment situation, a large number of new occupations have emerged, leading to job diversification. Digital economy has been widely concerned, but the development situation of new occupation practitioners has been ignored. At present, the overall development status of new occupations in China and the structural characteristics of new occupation practitioners in different cities are unclear."

[Comment 2] Abstract: The abstract of the original text only explains the method of data acquisition, but does not explain the source of data. The author should supplement this part.

[Response] Thank you for pointing out this problem in the manuscript. We have modified it in accordance with your comments, as shown below.

" The relevant data were obtained by mining big data on the recruitment platform and conducting a questionnaire survey on new professional practitioners, so as to compare and analyze the development status of new occupation practitioners in 4 first-tier cities and 15 new first-tier cities in China. "

[Comment 3] Abstract: The enlightenment and significance in the abstract should be more clear and have specific significance.

[Response]Thank you for your valuable advice. According to your suggestion, we have made a careful revision, as shown below.

Line 32 - Line 38: " Finally, we put forward the following countermeasures and suggestions: (1) Pay attention to the cultivation of new occupation practitioners. (2) Establish a classified database of new occupation talents. (3) Implementing the talent industry agglomeration strategy. (4) Paying attention to the coordinated development of new occupation practitioners in cities."

[Comment 4] The author can add the word "questionnaire survey" to the keyword.

[Response] Thank you for your reminder. We are very much agree with you. We have added the word " questionnaire survey " to the keyword.

[Comment 5] Introduction: In the original text “Chinese "14th Five-Year Plan" points out that we have to accelerate the digitization process, develop digital economy, promote a digital industrialization and industrial digitization, promote the deep integration between the digital economy and the real economy, and create an internationally competitive digital industry cluster.” Relevant documents of the policy should be supplemented here and added to the references.

[Response] Thank you for your reminder. We have made changes as you suggested, as shown below.

" Chinese "14th Five-Year Plan" points out that we have to accelerate the digitization process, develop digital economy, promote a digital industrialization and industrial digitization, promote the deep integration between the digital economy and the real economy, and create an internationally competitive digital industry cluster (2021). "

" References: The 14th five-year Plan for National Economic and Social Development of the people's Republic of China and the outline of the long-term objectives for 2035, 2021. The Central People's Government of the People's Republic of China. http://www.gov.cn/xinwen/2021-03/13/content_5592681.htm. Accessed 10 June 2022."

[Comment 6] Introduction: The theoretical basis and research content of this study are not described in detail in the original text. The author should make a detailed supplement.

[Response] Thank you for your valuable advice. We have made changes as you suggested, as shown below.

"The digital economy has been fundamentally reshaping the traditional industry structures and occupation paths around the world (Idowu & Elbanna, 2020). The new occupation practitioners are very important to the development of digital economy. Therefore, the qualitative and quantitative research on new occupation practitioners is very important. This paper constructed a development index system of new occupations practitioners from five dimensions (group size, cultural appreciation, salary level, occupation perception, and environmental perception). And the relevant data were obtained by using big data mining and questionnaire survey. The purpose of this paper is to explore the current situation and existing problems of the new occupation practitioners in major cities of China by comparing the development level of the new occupation practitioners in major cities of China. The study has two main contributions. First, the new occupation practitioners index system is constructed to measure the development level of the new occupation practitioners. Second, through the comparative analysis among cities, find out the existing problems, and put forward to promote the sustainable development of new occupation practitioners’ countermeasures and suggestions. "

[Comment 7] Introduction: The length of this part is large, and there is no certain correlation between the previous and subsequent paragraphs, which will give people the feeling of unclear logic and disorder. It is suggested that the author sort out the introduction by establishing a secondary title.

[Response] Thank you for your valuable advice. According to your suggestion, we have rearranged the logic of the introduction according to the questions you mentioned, and deleted and adjusted some of the contents. And we add part of the content to make the introduction more logical. And we have made changes as you suggested, marked in blue as shown in section 1.

[Comment 8] Literature review: In the literature review section, the author should write around the scientific problems mentioned above, so that this part has a clear logical line.

[Response] Thank you for your valuable advice. According to the questions you mentioned, we re-combed the literature review part. It is found that some of the contents are not very relevant to the research topic, and the whole logic is not clear enough. We deleted these contents. In order to make the logic of this part clearer, we have added a secondary title, as shown below.

2.1 new economic

2.2 new occupation

[Comment 9] Literature review: The author can summarize the literature review part by list.

[Response] Thank you very much for your comments. It was only in 2019 that China released a list of new occupations for the first time. We have not found any strongly relevant literature on the comprehensive evaluation of new professional practitioners. Therefore, we did not summarize the literature review part by list. And you may think that this part of the logic is not clear. We carefully read the literature review part, we modified its content, and added a second-level title to make the logic of this part clearer.

[Comment 10] Method: The part of "index system construction" in the original text should be one of the important parts of this paper. The references cited in this part are documents of earlier periods. The author can supplement the relevant authoritative documents in recent years.

[Response] Thank you for your valuable advice. According to your suggestion, we have added authoritative literature in recent years to make this part more evidence-based. The details are as follows.

" The evaluation index system refers to an organic whole with internal structure, which is composed of a number of indicators that characterize the characteristics of the evaluation object and their interrelations. It is widely used in various fields, such as green mining (Zhou et al., 2020), water problems (Chen et al., 2020), human competence (Li et al., 2020) and so on. By sorting out relevant literature, Zhao (2003) deemed that an industry's professionalism was composed of five factors: systematic knowledge system, professional moral creed, judgment standard, culture and social recognition. Professionalism is supported by a set of professional values, attitudes and behaviors (Grus et al., 2018; JHA et al., 2015). Tong (2018) proposed that the evaluation criteria of China's new-type professional farmers is supposed to include five indicators: professional cultural accomplishment, salary, skills, behavior norms and recognition. Following the characteristics of the new occupation group, we determine five first-level indicators in the evaluation index system, which are group size, cultural appreciation, salary level, occupation perception, and environmental perception.

The most basic premise of the formation of an occupation is that the profession reaches a certain scale. Therefore, occupation development is inseparable from the development of occupation size. Professional cultural appreciation mainly refers to the level of cultural knowledge of practitioners, which is the basic quality for practitioners to adapt and develop in their occupation (Ruiz-Lozano et al., 2019). Salary is generally divided into skill-based compensation (James O., et al., 2021), position-based compensation and competency-based compensation. Professional salary is one of the main influencing factors of occupation differentiation and social division of labor. Occupation perception includes recognition and satisfaction. The recognition and satisfaction of a profession are closely related to its social status, working environment (Bjork et al., 2019), work intensity, income level, vocational welfare, development space, professional characteristics and so on. 

[Comment 11] Method: In the "data processing and model construction" part of the original text, the author should supplement the references of using this data processing method, and please explain the reasons and advent ages of using this data processing method in combination with the content of this article.

[Response] Thank you for your valuable advice. We have added relevant content in accordance with your suggestion, as shown below.

" Analytic hierarchy process (AHP) (Saaty, 2013) is a systematic and hierarchical analysis method that combines qualitative and quantitative analysis. The characteristic of this method is that on the basis of in-depth study of the nature, influencing factors and internal relations of complex decision-making problems, the thinking process of decision-making is mathematized by using less quantitative information. thus, it provides a simple decision-making method for complex decision-making problems with multi-objectives, multi-criteria or no structural characteristics. The specific steps are as follows."

" In order to eliminate the impact of diverse dimensions on the results, the range method is employed and the results are controlled within [60,100] (Lyu et al., 2020). "

" Considering the characteristics of the index and the convenience of operation, the weighted arithmetic mean model is applied to synthesize the index (Aruchunan et al., 2021). "

[Comment 12] Method: In the "data source" part of the original text, the author should supplement the relevant websites and database links of the data source. At the same time, the author should supplement the questionnaire and explain the reliability and validity of the questionnaire.

[Response] Thank you very much for your valuable advice. According to your suggestion, we have supplemented the links to the relevant websites and databases. The reliability and validity are mainly aimed at the questionnaire of the scale, our questionnaire only investigates the objective reality, which is mainly based on explicit variables. It is not suitable for reliability and validity. So we do not explain the reliability and validity of the questionnaire.Thank you again for your valuable advice.

[Comment 13] Method: Empirical analyses

[Comment 14] Method: The author should explain the software used in the empirical study.

[Response] Thank you for your valuable advice. Add relevant content to our manuscript, marked in blue as shown in line 316.

[Comment 15] Method: The author should place Table 4 on the same page.

[Response]Thank you for raising this question. We have adjusted Table 4 to the same page.

[Comment 16] The author can supplement a Discussion section to compare the results of this paper with those of predecessors.

[Response]Thank you very much for your valuable advice. What you mentioned is really very important. We didn't do this work because there was too little related research. It was only in 2019 that China released a list of new occupations for the first time. The topic is new. If we write this part, we must compare it with the less relevant literature, which we think is of little help to improve the quality of our articles. Thank you again for your suggestion, if you think it is necessary, we will supplement this part of the work.

[Comment 17] Conclusions: The research purpose, content and methods of the article should be described in the conclusion of the original text.

[Response]Thank you for your valuable advice. We have revised it carefully in accordance with your opinion. The relevant content is shown below.

In order to better evaluate the development status of the new occupation practitioners in China and discover the relevant issues during the development. So as to provide reference for the development of digital economy. This paper constructed an index system of new occupation practitioners from five dimensions, including group size, cultural literacy, salary level, occupation perception and environmental perception. Based on the relevant data obtained from big data mining and questionnaire survey, the current state of development of new occupational practitioners in 19 front-line cities in China was measured. 

[Comment 18] Conclusions: When expressing the research results, the language can be appropriately simplified. At the same time, the enlightenment given by the author should go beyond the Chinese perspective.

[Response]Thank you for your comments. We very much agree with the problem you mentioned. We have added a paragraph to break through China's vision. The details are as follows.

" Not only China, but also the digital economy has a profound impact on all countries in the world. The development of new occupation practitioners is related to the development of digital economy. Countries all over the world should be clearly aware of the current situation of the development of new occupation practitioners, so as to regulate and control the healthy development of the digital economy."

Response to Reviewer #2:

[General Comment] This is an interesting manuscript that examines the emergence of a number of new professions in China and how they have developed in 19 first tier cities.

[Response] Thank you very much for your recognition of my work and research, I am so glad to receive your recommendation for my paper.

[Comment 1] In the first line of the abstract, you mention the complex economic environment and severe employment situation, but this doesn't really seem to come through in the body of your work.

[Response] Thank you for your valuable advice. According to the question you mentioned, I added the background of the severe employment situation in the introduction.

"Under the epidemic of COVID-19, in 2020, our country's employment situation is serious. According to the Department of human resources and social security, the country's newly employed population decreased year-on-year by 950000 in the first quarter. In such an environment, the new occupation such a flexible and boundless occupation becomes the key to economic recovery in our country. "

[Comment 2] I found the introduction and literature review to lack coherence - I wasn't always sure of what the 'story' you are trying to tell is. One suggestion is to use sub-headings to help organise your thinking and writing in a logical way. At the end of the introduction you should state your clear aims/objectives/research questions/hypotheses for this work.

[Response]Thank you for your valuable advice. We have rearranged the logic of the introduction according to the questions you mentioned, and deleted and adjusted some of the contents. And we add part of the content to make the introduction more logical. At the same time, we sort out the literature review. As you suggested, we used subheadings to make the logic of this part clearer, and deleted some irrelevant content. And we have made changes as you suggested, marked in blue as shown in section 1 and section 2.

[Comment 3] What are 'first tier cities'? an international audience wouldn't be likely to understand what they are without some explanation.

[Response]Thank you very much for your advice. In the 3.2.1 part of the article, we introduce the first-tier cities. According to your suggestion, we have supplemented this part. As shown below.

The first-tier city refers to the metropolis which plays an important role and plays a leading role in the national political, economic and other social activities. This article takes the new occupation practitioners in 19 “first-tier” cities in China as the research object. The 19 first-tier cities are shown in Table 1. 

[Comment 4] It might be worth introducing things in a reverse chronological order - what started the "new economy" and how has it progressed to the digital economy and beyond? What are the key features of each of these 'phases'?

[Response] Thank you very much for your valuable comments on our paper. This is a very good opinion, and we have made appropriate changes to the paper in the light of your comments. The focus of our paper is on new occupation practitioners. The "new economy" is only as the background, so we do not spend too much on the "new economy". Thank you again for your valuable advice.

[Comment 5] Line 128 - five new what?

[Response] Thank you very much for your valuable comments on our paper. It includes new products, new services, new industries, new forms of business and new models. However, according to your previous suggestion, we found that these contents are not highly compatible with our article, so we deleted them.

[Comment 6] An example at line 129 - 'economy is developing in full swing' - this is very jargon/colloquial and should be re-written. The audience in PLOS One is broad and won't necessarily know or understand the jargon. There are instances of this throughout the manuscript.

[Response] Thank you for your comments. We deleted it to make it more in line with an academic paper.

[Comment 7] Section 3.1.1 - I would suggest starting with a brief explanation of what the index system is, why it is needed/important and how it works (at a high level). Then get into the details of your index.

[Response] Thank you for your valuable advice. According to your suggestion, we have revised it carefully. The details are as follows.

" The evaluation index system refers to an organic whole with internal structure, which is composed of a number of indicators that characterize the characteristics of the evaluation object and their interrelations. It is widely used in various fields, such as green mining (Zhou et al., 2020), water problems (Chen et al., 2020), human competence (Li et al., 2020) and so on. By sorting out relevant literature, Zhao (2003) deemed that an industry's professionalism was composed of five factors: systematic knowledge system, professional moral creed, judgment standard, culture and social recognition. Professionalism is supported by a set of professional values, attitudes and behaviors (Grus et al., 2018; JHA et al., 2015). Tong (2018) proposed that the evaluation criteria of China's new-type professional farmers is supposed to include five indicators: professional cultural accomplishment, salary, skills, behavior norms and recognition. Following the characteristics of the new occupation group, we determine five first-level indicators in the evaluation index system, which are group size, cultural appreciation, salary level, occupation perception, and environmental perception."

[Comment 8] Lines 208-227 - this is some good background/context to the index, however, there are no references. Please reference this section appropriately.

[Response] Thank you for your affirmation of our work. It is our mistake to have such a problem. We have supplemented the references according to your opinion. As shown below.

The most basic premise of the formation of an occupation is that the profession reaches a certain scale. Therefore, occupation development is inseparable from the development of occupation size. Professional cultural appreciation mainly refers to the level of cultural knowledge of practitioners, which is the basic quality for practitioners to adapt and develop in their occupation (Ruiz-Lozano et al., 2019). Salary is generally divided into skill-based compensation (James O., et al., 2021), position-based compensation and competency-based compensation. Professional salary is one of the main influencing factors of occupation differentiation and social division of labor. Occupation perception includes recognition and satisfaction. The recognition and satisfaction of a profession are closely related to its social status, working environment (Bjork et al., 2019), work intensity, income level, vocational welfare, development space, professional characteristics and so on.

[Comment 9] At line 271, what are the Liepin and Ganji networks? What data do they hold? How were they mined? What are they?

[Response] Thank you for mentioning the ambiguity of our article. We have made a supplementary explanation in this part. As shown below.

"(1) Big data mining. Liepin and Ganji is a famous job recruitment platform in China. They hold data on the demand for new occupation in the market. By big data mining of the whole network, such as Liepin, Ganji, etc., obtained the annual demand information of new occupation practitioners in 19 “first-tier” cities since the emergence of a large number of new jobs, and obtained a total of 7.14 million data in 2020. "

[Comment 10] Lines 286-288 - report the demographics in a consistent way - either by age or by a birth year range, not a mix of the two.

[Response] Thank you for your advice. We modified it. As shown below.

" As for age, people aged between 18 and 24 take up 32.60%, people aged between 25 and 29 take up 31.72%, and the proportion of people under the age of 40 is 92.87%."

[Comment 11] Overall, the manuscript requires a thorough English language proofing; this will make it easier to assess its academic merits.

[Response] Thank you very much for your help in our paper. In response to the opinions mentioned above, we are already looking for relevant institutions to improve the quality of the English language. Thank you again for your help in our paper.

---

## [Decision Letter · Decision Letter 1]

7 Sep 2022

PONE-D-22-01167R1Comparative study on the development of new occupation practitioners in China's “first-tier” citiesPLOS ONE

Dear Dr. Chen,

Thank you for submitting your manuscript to PLOS ONE. After careful consideration, we feel that it has merit but does not fully meet PLOS ONE’s publication criteria as it currently stands. Therefore, we invite you to submit a revised version of the manuscript that addresses the points raised during the review process.

We look forward to receiving your revised manuscript.

Kind regards,

Ziqiang Zeng, Ph.D.

Academic Editor

PLOS ONE

Journal Requirements:

Reviewers' comments:

Reviewer's Responses to Questions

**Comments to the Author**

1. If the authors have adequately addressed your comments raised in a previous round of review and you feel that this manuscript is now acceptable for publication, you may indicate that here to bypass the “Comments to the Author” section, enter your conflict of interest statement in the “Confidential to Editor” section, and submit your "Accept" recommendation.

Reviewer #1: (No Response)

Reviewer #2: All comments have been addressed

2. Is the manuscript technically sound, and do the data support the conclusions?

Reviewer #1: Yes

Reviewer #2: (No Response)

3. Has the statistical analysis been performed appropriately and rigorously? 

Reviewer #1: Yes

Reviewer #2: (No Response)

4. Have the authors made all data underlying the findings in their manuscript fully available?

Reviewer #1: No

Reviewer #2: (No Response)

5. Is the manuscript presented in an intelligible fashion and written in standard English?

Reviewer #1: No

Reviewer #2: (No Response)

6. Review Comments to the Author

Reviewer #1: 1. The current literature review is mostly from Chinese authors, which is very lacking in international perspective. Please revisit the more diverse international literature and strengthen the critical literature review.

2. After reporting your results, compare your results with similar studies. Maybe a discussion section could be added.

3. At present, the language of this manuscript is not standardized and fluent. Please make sure this manuscript has been checked by a native English-speaking professor.

Reviewer #2: Thank you for your comprehensive revisions.

I feel that you have addressed the reviewer comments sufficiently.

My only substantive comment is that the whole manuscript needs to be language and grammar proofed by a native English speaker before publication.

Please check your headings and sub-headings for grammatical soundness - for example, "2.1 New economic" doesn't sound right - should it be "New economy"?

7. PLOS authors have the option to publish the peer review history of their article (what does this mean?). If published, this will include your full peer review and any attached files.

Reviewer #1: No

Reviewer #2: No

---

## [Author Response · Author response to Decision Letter 1]

30 Sep 2022

Dear editors and reviewers,

We would like to thank you and the two reviewers and editors for their valuable comments and kind suggestions regarding the manuscript we submitted. These comments and suggestions directed us to improve the content of our study. We have revised our paper again, and the revised version of the manuscript incorporates the comments and suggestions of the reviewers. 

The details of modification were listed point by point:

Response to Reviewer #1:

[Comment 1] The current literature review is mostly from Chinese authors, which is very lacking in international perspective. Please revisit the more diverse international literature and strengthen the critical literature review.

[Response] Thank you very much for your comments, which is highly appreciated. In the new occupation section of the literature review, we have also added relevant research by authors from other countries and, at the same time, added critical perspectives.

“Initially, the United States was the country with the best new career development because its culture emphasized independence and self-realization, and its labor market practices were quite flexible and open (Pringle and Mallon，2003；Mayhofer et al.，2004). But today's society is becoming more and more "mobile" and uncertain (Bauman, 2007). Other countries are also interested in this (Druker and Stanworth 2004).”

“Such a survey is important because it is an important source of job satisfaction (Dawis and Lofquist, 1984).”

[Comment 2] After reporting your results, compare your results with similar studies. Maybe a discussion section could be added.

[Response] Thank you for pointing out this problem in the manuscript. We have added a discussion section to compare it with the research of other scholars. First of all, this study is summarized. Then, for some research conclusions, we compared with the previous related research. This highlights the significance of our research. Thank you again for your valuable advice. The detailed discussion is as follows.

“To better evaluate the development status of new occupation practitioners in China, discover relevant issues during development, and provide a reference for the development of a digital economy, this study measured the development level of new occupation practitioners in 19 “first-tier” cities in China from five dimensions: group size, cultural literacy, salary level, occupation perception, and environmental perception. Based on a comparative analysis of 19 “first-tier” cities, this study describes the development level of new occupation practitioners in China. The results indicate that the new occupation practitioners have a high degree of recognition of and satisfaction with the new occupations and hold a positive and optimistic attitude toward the development prospects of the new occupation practitioners. The four first-tier cities lead 15 new first-tier cities in the development level of new occupation practitioners. Among the 15 new first-tier cities, Chengdu and Hangzhou did well in the development of new occupation practitioners. Comparatively, other new first-tier cities have a relatively slow start and late development of new occupations. In most cities, the development of new occupation practitioners is not coordinated with the five indicators, severely affecting the healthy development of new occupation practitioners. The policy strength of relevant departments in some cities is low and has not taken the shape of a good policy environment and social environment to promote the sustainable development of new occupation practitioners.

In this study, most new occupations are borderless. Their work is very flexible. For example, online delivery staff and ride-hailing drivers can work according to their own time. There are fewer restrictions on working hours and workplaces. They are more likely to have multiple jobs (Sprajcer, 2021), and at the same time, they tend to be younger (Sprajcer, 2021); this is consistent with our research results. New occupation practitioners are more satisfied with their careers, and Kundi (2020) showed that this kind of career practitioner has higher satisfaction.

The results show that new occupation practitioners in the four first-tier cities have a higher level of development. Cities with better economic development are more conducive to developing new occupations. The new occupation is the product of conforming to the development of the times, and cities with better economic development can provide a more favorable environment for the development of the new occupation. A long time ago, some scholars proposed that borderlessness would soon become the main career model (Arthur and Rousseau, 1996), which has become a reality. In recent years, its scale has expanded. An increasing number of people benefit from such careers (Handy, 1989).”

[Comment 3] At present, the language of this manuscript is not standardized and fluent. Please make sure this manuscript has been checked by a native English-speaking professor.

[Response] Thank you for your valuable advice. This article has been proofread by a professional body for language and grammar. The proof material is shown in system.

Response to Reviewer #2:

[Comment 1] I feel that you have addressed the reviewer comments sufficiently.My only substantive comment is that the whole manuscript needs to be language and grammar proofed by a native English speaker before publication. Please check your headings and sub-headings for grammatical soundness - for example, "2.1 New economic" doesn't sound right - should it be "New economy"?

[Response] Thank you for your valuable advice. We have corrected the problem you mentioned above. This article has been proofread by a professional body for language and grammar. The proof material is shown in system.

---

## [Decision Letter · Decision Letter 2]

31 Oct 2022

PONE-D-22-01167R2Comparative study on the development of new occupation practitioners in China's “first-tier” citiesPLOS ONE

Dear Dr. Zhang,

Thank you for submitting your manuscript to PLOS ONE. After careful consideration, we feel that it has merit but does not fully meet PLOS ONE’s publication criteria as it currently stands. Therefore, we invite you to submit a revised version of the manuscript that addresses the points raised during the review process.

We look forward to receiving your revised manuscript.

Kind regards,

Ziqiang Zeng, Ph.D.

Academic Editor

PLOS ONE

Reviewers' comments:

Reviewer's Responses to Questions

**Comments to the Author**

1. If the authors have adequately addressed your comments raised in a previous round of review and you feel that this manuscript is now acceptable for publication, you may indicate that here to bypass the “Comments to the Author” section, enter your conflict of interest statement in the “Confidential to Editor” section, and submit your "Accept" recommendation.

Reviewer #1: (No Response)

Reviewer #2: All comments have been addressed

2. Is the manuscript technically sound, and do the data support the conclusions?

Reviewer #1: Yes

Reviewer #2: (No Response)

3. Has the statistical analysis been performed appropriately and rigorously? 

Reviewer #1: Yes

Reviewer #2: (No Response)

4. Have the authors made all data underlying the findings in their manuscript fully available?

Reviewer #1: Yes

Reviewer #2: (No Response)

5. Is the manuscript presented in an intelligible fashion and written in standard English?

Reviewer #1: No

Reviewer #2: (No Response)

6. Review Comments to the Author

Reviewer #1: The authors' attitude toward revision is unsatisfactory.

The manuscript lacks a critical review of the latest literature published in 2022.

The language of this manuscript is not standard and fluent.

There are some spelling mistakes that have not been corrected. Please check the upper and lower case of each section.

Conclusions and discussion should be split into 2 sections.

Reviewer #2: The authors appear to have responded to the feedback.

Please make sure that you have a good language proof of the manuscript

7. PLOS authors have the option to publish the peer review history of their article (what does this mean?). If published, this will include your full peer review and any attached files.

Reviewer #1: No

Reviewer #2: No

---

## [Author Response · Author response to Decision Letter 2]

11 Nov 2022

Dear editors and reviewers，

We would like to thank you and the two reviewers and editors for their valuable comments and kind suggestions regarding the manuscript we submitted. These comments and suggestions directed us to improve the content of our study. We have revised our paper substantially, and the revised version of the manuscript incorporates the comments and suggestions of the reviewers. 

The details of modification were listed point by point:

Response to Reviewer #1:

Question 1

1. The manuscript lacks a critical review of the latest literature published in 2022.

Response:

Thank you for your valuable advice. According to your suggestion, we have added some relevant latest literature published in 2022 in section 2. Part of the added literature is shown below.

[23] Wang, Y., Wei YD., & Sun, BD., 2022. New economy and national city size distribution. Habitat International, 127, 102632.

[24] Nham, NTH., & Ha L., 2022. Making the circular economy digital or the digital economy circular? Empirical evidence from the European region. Technology in Society, 70: 102023.

[25] Li, ZG., & Wang J., 2022. The Dynamic Impact of Digital Economy on Carbon Emission Reduction: Evidence City-level Empirical Data in China. Journal of Cleaner Production, 351: 131570.

[28] Lederman, D., & Zouaidi, M., 2022. Incidence of the Digital Economy and Frictional Unemployment: International Evidence. Applied Economics, 54(51): 5873-5888.

[45] Mousa, M., & Chaouali, W., 2022. Job crafting, meaningfulness and affective commitment by gig workers towards crowdsourcing platforms. Personnel Review. DOI: 10.1108/PR-07-2021-0495.

Question 2

2. The language of this manuscript is not standard and fluent. There are some spelling mistakes that have not been corrected. Please check the upper and lower case of each section.

Response:

Thank you for raising this question. According to your suggestion, We have carefully checked the full text for spelling and grammar mistakes. And we would like to thank Editage (www.editage.cn) for English language editing. Please see the attachment for the proof.

Question 3

3. Conclusions and discussion should be split into 2 sections.

Response:

Thank you for your valuable advice. According to your suggestion, the conclusion and discussion have been divided into two separate sections.

Response to Reviewer #2:

The authors appear to have responded to the feedback.

Please make sure that you have a good language proof of the manuscript

Response:

Thank you for your valuable advice. We have carefully checked the full text for spelling and grammar mistakes. And we would like to thank Editage (www.editage.cn) for English language editing. Please see the attachment for the proof.

Sincerely thanks again to the editor and reviewers for their valuable comments on this version of this paper.

---

## [Decision Letter · Decision Letter 3]

13 Feb 2023

PONE-D-22-01167R3Comparative study on the development of new occupation practitioners in China's “first-tier” citiesPLOS ONE

Dear Dr. Zhang,

Thank you for submitting your manuscript to PLOS ONE. After careful consideration, we feel that it has merit but does not fully meet PLOS ONE’s publication criteria as it currently stands. Therefore, we invite you to submit a revised version of the manuscript that addresses the points raised during the review process.

We look forward to receiving your revised manuscript.

Kind regards,

Ziqiang Zeng, Ph.D.

Academic Editor

PLOS ONE

Journal Requirements:

Reviewers' comments:

Reviewer's Responses to Questions

**Comments to the Author**

1. If the authors have adequately addressed your comments raised in a previous round of review and you feel that this manuscript is now acceptable for publication, you may indicate that here to bypass the “Comments to the Author” section, enter your conflict of interest statement in the “Confidential to Editor” section, and submit your "Accept" recommendation.

Reviewer #1: All comments have been addressed

Reviewer #3: (No Response)

2. Is the manuscript technically sound, and do the data support the conclusions?

Reviewer #1: Yes

Reviewer #3: Yes

3. Has the statistical analysis been performed appropriately and rigorously? 

Reviewer #1: Yes

Reviewer #3: Yes

4. Have the authors made all data underlying the findings in their manuscript fully available?

Reviewer #1: Yes

Reviewer #3: Yes

5. Is the manuscript presented in an intelligible fashion and written in standard English?

Reviewer #1: Yes

Reviewer #3: No

6. Review Comments to the Author

Reviewer #1: After reading the revised version, I am quite satisfied. The authors have carefully revised the manuscript and the present version is acceptable.

Reviewer #3: Dear authors,

I have reviewed the manuscript and fond that it has improved according to the review comments. As a new involved reviewer, I have some advice for minor revision.

(1) your references are somehow old, and I can even find some in the last century. I suggest you update your references. For instance, in Line 49-51, you can refer to the following papers: Research Progress of Green Marketing in Sustainable Consumption based on CiteSpace Analysis; Line 128: Digital economy is of great significance to carbon emission reduction[X]; X: Bibliometric review of carbon neutrality with CiteSpace: evolution, trends, and framework; 47, 60, 61 reference should be changed; they are too old.

(2) you had better look for a professional language editing service (such as AJE or other PLOS certified services) to enhance your language. There are so many Chinglish and redundent expressions.

(3) Please redraw figure 1: current version is not clear (I am not sure whether because of the system's compression, or it is actually not clear). Maybe you can add different colors in different dimensions to make it more beautiful.

7. PLOS authors have the option to publish the peer review history of their article (what does this mean?). If published, this will include your full peer review and any attached files.

Reviewer #1: No

Reviewer #3: No

---

## [Author Response · Author response to Decision Letter 3]

20 Mar 2023

Dear editors and reviewers,

Thank you very much for your valuable comments and kind suggestions regarding our research paper. Each of your feedback to strengthen our manuscript and we corrected point by point the manuscript accordingly. Your comments are in bold text and our responses in plain italics. The revised version contains your comments and suggestions, as well as recorded the changed below.

Response to Reviewer #1:

[Comment 1] After reading the revised version, I am quite satisfied. The authors have carefully revised the manuscript and the present version is acceptable.

[Response] First of all, thank you very much for your affirmation of our work. Thank you again for your valuable suggestions in the manuscript review, which has greatly improved the quality of our paper.

Response to Reviewer #3:

[Comment 1] Your references are somehow old, and I can even find some in the last century. I suggest you update your references. For instance, in Line 49-51, you can refer to the following papers: Research Progress of Green Marketing in Sustainable Consumption based on CiteSpace Analysis; Line 128: Digital economy is of great significance to carbon emission reduction[X]; X: Bibliometric review of carbon neutrality with CiteSpace: evolution, trends, and framework; 47, 60, 61 reference should be changed; they are too old.

[Response] Thanks for all your comments and suggestions. We have made your recommended changes. According to your proposed, we cited two papers you mentioned and updated the relatively old references.

“[4] Geng, YQ. & Maimaituerxun, M., 2022. Research Progress of Green Marketing in Sustainable Consumption based on CiteSpace Analysis. SAGE Open, 12(3), 1-19.”

“[26] Geng, YQ., Zhu, RJ. & Maimaituerxun, M., 2022. Bibliometric review of carbon neutrality with CiteSpace: evolution, trends, and framework. Environmental Science and Pollution Research, 29(51), 76668-76686.”

” [49] Yukl, G., Mahsud,R., Prussia, G. & Hassan, S., 2019. Effectiveness of broad and specific leadership behaviors. Personnel Review, 48(3), 774-783.”

 “[62] Inkson, K., Gunz, H., Ganesh, S. & Roper, J., 2012. Boundaryless Careers: Bringing Back Boundaries. Organization Studies, 33(3), 323-340.”

“[63] Kundi, YM., Hollet-Haudebert, S. & Peterson, J., 2021. Linking Protean and Boundaryless Career Attitudes to Subjective Career Success: A Serial Mediation Model. Journal of Career Assessment, 29(2), 263-282.”

[Comment 2] You had better look for a professional language editing service (such as AJE or other PLOS certified services) to enhance your language. There are so many Chinglish and redundant expressions.

[Response] Thank you for your valuable advice. We have corrected the problem you mentioned above. This article has proofread by a professional body for language and grammar. We revised the introduction、finding、discussion and conclusion.

[Comment 3] Please redraw figure 1: current version is not clear (I am not sure whether because of the system's compression, or it is actually not clear). Maybe you can add different colors in different dimensions to make it more beautiful.

[Response] Thank you for your kind suggestions. As you mentioned, the old one is not clear, we added other colors to differentiate. As shown in Figure1.

---

## [Editor Report · Decision Letter 4]

27 Mar 2023

The development of new occupation practitioners in China’s first-tier cities: a comparative analysis

PONE-D-22-01167R4

Dear Dr. Zhang,

We’re pleased to inform you that your manuscript has been judged scientifically suitable for publication and will be formally accepted for publication once it meets all outstanding technical requirements.

Kind regards,

Ziqiang Zeng, Ph.D.

Academic Editor

PLOS ONE
---

## [Editor Report · Acceptance letter]

16 Oct 2023

PONE-D-22-01167R4 

The development of new occupation practitioners in China’s first-tier cities: a comparative analysis 

Dear Dr. Zhang:

I'm pleased to inform you that your manuscript has been deemed suitable for publication in PLOS ONE. Congratulations! Your manuscript is now with our production department. 

Kind regards, 

on behalf of

Dr. Ziqiang Zeng 

Academic Editor

PLOS ONE